# Diazotroph *Paenibacillus triticisoli* BJ-18 Drives the Variation in Bacterial, Diazotrophic and Fungal Communities in the Rhizosphere and Root/Shoot Endosphere of Maize

**DOI:** 10.3390/ijms22031460

**Published:** 2021-02-02

**Authors:** Yongbin Li, Qin Li, Sanfeng Chen

**Affiliations:** State Key Laboratory for Agrobiotechnology and College of Biological Sciences, China Agricultural University, Beijing 100094, China; ybli@cau.edu.cn (Y.L.); lqliqin1@126.com (Q.L.)

**Keywords:** bacterial community, diazotrophic community, fungal community, diazotroph, *Paenibacillus*

## Abstract

Application of diazotrophs (N_2_-fixing microorganisms) can decrease the overuse of nitrogen (N) fertilizer. Until now, there are few studies on the effects of diazotroph application on microbial communities of major crops. In this study, the diazotrophic and endospore-forming *Paenibacillus triticisoli* BJ-18 was inoculated into maize soils containing different N levels. The effects of inoculation on the composition and abundance of the bacterial, diazotrophic and fungal communities in the rhizosphere and root/shoot endosphere of maize were evaluated by sequencing the 16S rRNA, *nifH* gene and ITS (Inter Transcribed Spacer) region. *P. triticisoli* BJ-18 survived and propagated in all the compartments of the maize rhizosphere, root and shoot. The abundances and diversities of the bacterial and diazotrophic communities in the rhizosphere were significantly higher than in both root and shoot endospheres. Each compartment of the rhizosphere, root and shoot had its specific bacterial and diazotrophic communities. Our results showed that inoculation reshaped the structures of the bacterial, diazotrophic and fungal communities in the maize rhizosphere and endosphere. Inoculation reduced the interactions of the bacteria and diazotrophs in the rhizosphere and endosphere, while it increased the fungal interactions. After inoculation, the abundances of *Pseudomonas*, *Bacillus* and *Paenibacillus* in all three compartments, *Klebsiella* in the rhizosphere and *Paenibacillus* in the root and shoot were significantly increased, while the abundances of *Fusarium* and *Giberella* were greatly reduced. *Paenibacillus* was significantly correlated with plant dry weight, nitrogenase, N_2_-fixing rate, P solubilization and other properties of the soil and plant.

## 1. Introduction

Plants host distinct microbial communities on and inside their tissues, designated as the plant microbiome [1,2,3]. The microorganisms in the plant microbiome could be beneficial, harmful or neutral, according to their effects on plant development [4]. Rhizobacteria in the plant rhizosphere are more active in this zone and possess diverse metabolic capabilities and thus play a crucial role in plant health and soil fertility [5]. Many rhizobacteria are PGRB (plant growth-promoting rhizobacteria), inhabiting around or on the root surface, and may directly or indirectly promote plant growth and development in the vicinity of the rhizosphere [6,7]. Endophytes that live inside plant tissues for at least part of their life cycles do not cause harm and could establish a mutualistic association [8,9]. The mechanisms used by endophytic bacteria to promote plant growth are similar to PGPR. Furthermore, the endophytic bacteria can also provide plants with some advantages in resistance against drought and salinity [10,11]. PGPR and endophytic bacteria have great usage as agricultural inoculants, such as biofertilization and biocontrol of pathogens [12,13]. Commercialized PGPR strains mainly include members of the *Agrobacterium*, *Azospirillum*, *Bacillus*, *Paenibacillus*, *Pseudomonas* and *Streptomyces* [14,15].

It has been demonstrated that plant bacterial endophytes are mainly recruited from soil, which then migrate to stems and leaves via the apoplast in xylem vessels [16,17,18]. Thus, microbiomes in the plant leaf/shoot endosphere have significant overlaps with those in roots [19,20,21]. The abundance and diversity of the bacteria in the root and leaf endospheres are significantly less diverse than those in the rhizosphere and bulk soil [22]. The number of bacterial cells in bulk soil is about 10^6^–10^9^ bacterial cells/g soil and in the rhizosphere about 10^6^–10^9^ bacterial cells/g soil), while the number of bacterial cells within the root endosphere is about 10^4^–10^8^ per gram of root tissue [10,23]. This suggest that plant roots act as “gatekeepers” to screen soil bacteria that enter root and leaf tissues from the rhizosphere [10].

Soil is one of the richest microbial ecosystems on earth [24]. The specificity of the microbiomes in the plant rhizosphere is modulated by the complex interactions between plant hosts and soil type [25]. Each plant species has its specific microbial community [26,27,28]. Plant root exudates are a driving force in this process [29,30,31,32], since plant roots secrete a wide range of compounds, including organic acid ions, inorganic ions, sugars, vitamins, amino acids and siderophores [33]. For example, dicarboxylates in tomato root exudates favored the growth of *Pseudomonas* biocontrol strains [34,35] and legumes secreted flavonoids to establish symbioses with N_2_-fixing rhizobia [36]. Benzoxazinoids, a group of secondary metabolites present in maize, affected microbial communities in the rhizosphere and in roots/shoots, shown by analyzing different genotypes (benzoxazinoid knock-out mutants and their parental lines) [37]. Different plant organs are also associated with different endophytic microbial communities [23]. Edwards et al. [27] analyzed the bacterial and archaeal microbiomes from three root-associated compartments of crop rice—the endosphere (root interior), rhizoplane (root surface) and rhizosphere (soil close to the root surface)—and found that each harbors a distinct microbiome; i.e., varying considerably in the composition of the bacterial community. Studies on the rhizosphere bacterial diversity of 27 modern maize inbreds grown under field conditions showed substantial variation in bacterial richness, diversity and relative abundances of taxa between bulk soil and the maize rhizosphere, as well as between fields [26]. Comparative analysis of the bacterial community structure in the maize rhizosphere and soil showed that PGPR, such as *Nonomuraea*, *Thiobacillus* and *Bradyrhizobium*, appeared to be more abundant in the rhizosphere than bulk soil, indicating that the rhizosphere has more impact on soil microorganisms [38]. Furthermore, application of the beneficial microbes reshaped the rhizosphere microbiome. For example, *Bacillus amyloliquefaciens*-enriched bio-fertilizer and *Trichoderma guizhouense*-enriched bio-fertilizer treatments significantly reduced the cumulative incidence of vanilla and banana *Fusarium* wilt diseases by reshaping the soil microbiome [39,40]. Inoculation with N_2_-fixing *Pseudomonas stutzeri* A1501 significantly altered the composition of the diazotrophic communities of the rhizosphere and root surface where *P. stutzeri* A1501 became dominant in the rhizosphere, and also increased the population of indigenous diazotrophs and ammonia oxidizers [41].

Biological nitrogen (N) fixation plays an important role in the N cycle. In addition to symbiotic N_2_-fixing rhizobia associated with legumes, the non-symbiotic diazotrophs were also important contributors to the N nutrition of non-legumes [18,42,43]. N_2_-fixing bacteria were mainly identified in soils or plant issues by analyzing the nitrogenase (*nifH*) genes through PCR [44]. The non-symbiotic diazotrophs are highly diverse and associated with plants in different ways. For example, *Azospirillum brasillense* and *Azotobacter* spp. are associative with the plant root, and *Azoarcus*, *Gluconacetobacter* and *Herbaspirillum* are endophytic diazotrophs living within root and shoot tissues [45,46]. Endophytic N_2_-fixing bacteria constitute only a small proportion of total endophytic bacteria [47,48]. Endophytic N_2_-fixing bacteria may have an advantage over associative diazotrophic bacteria and rhizobacteria, since they live within plant tissues where better niches are established for N_2_ fixation and assimilation of the fixed N_2_ by the plant [49,50]. Biological N fixation quantification experiments showed that associative and endophytic bacteria can fix N_2_ in plant tissues with higher efficiency [45]. For example, inoculation with endophytic *Gluconacetobacter diazotrophicus* enhanced sugarcane yield by providing 50–80% N from biological N fixation [51]. N_2_-fixing bacteria present in the mucilage of aerial roots contributed 29–82% of the N nutrition for Sierra Mixe maize [52]. Diazotrophs are generally considered as a subset of PGPR by N fixation. Research on diazotrophs has mainly focused on quantifying the amount of N_2_ fixed, identification of the diazotrophs and regulation mechanisms [45,46]. However, studies on whether diazotrophs can induce variation in the plant microbiome are very limited.

*Paenibacillus triticisoli* BJ-18 is a diazotrophic and endospore-forming species isolated from the wheat rhizosphere by our laboratory [53]. Recent studies have shown that *P. triticisoli* BJ-18 produced siderophores and indolic acids that play roles in plant growth promotion [53,54,55]. GFP-labelling showed that *P. triticisoli* BJ-18 could colonize on the plant root surface and inside the roots, stems and leaves of maize, wheat and cucumber to promote their growth, and ^15^N isotope enrichment analysis showed that this bacterium provided plants with 12.9–36.4% of their N by biological N fixation [18].

In this study, the *P. triticisoli* BJ-18 cells were inoculated into maize soils containing low N, medium N and high N levels of fertilizer, respectively, with non-inoculated treatments as controls. The population dynamics of the *P. triticisoli* BJ-18 cells in the rhizosphere and root/shoot endosphere of maize were investigated. The effects of the inoculation on the structures of the bacterial, diazotrophic and fungal communities in three compartments were evaluated by sequencing the 16S rRNA gene, ITS region and *nifH* gene coding for a subunit of nitrogenase. The relationships of the microbial communities with environmental factors were also investigated. Our study revealed that reshaping the microbiome structure is another major mechanism used by diazotrophs to promote plant growth.

## 2. Results

### 2.1. Population Dynamics of P. triticisoli BJ-18 and Copy Numbers of 16S rRNA, nifH Gene and ITS 

The population densities of *P. triticisoli* BJ-18 in the maize rhizosphere and root/shoot endosphere were determined by qPCR with the specific primers for the *nifH* gene (encoding a subunit of nitrogenase) of *P. triticisoli* BJ-18 during the plant growth period from Days 7 to 35. We found that *P. triticisoli* BJ-18 was specifically detected in the inoculated samples rather than non-inoculated samples, suggesting that the primers for the *nifH* gene were highly specific. The densities of *P. triticisoli* BJ-18 in all the rhizosphere, root and shoot samples increased firstly, then declined and finally persisted (Figure 1), suggesting that this diazotroph was able to multiply in the soil and plant tissues. The highest copy number of *P. triticisoli* BJ-18 reached 8.64 × 10^3^ per gram rhizosphere (Figure 1A), 5.80 × 10^2^ per gram root (Figure 1B) and 1.73 × 10^2^ per gram shoot (Figure 1C) under low N, suggesting that soil N status affected the population densities in the rhizosphere and root/shoot endosphere. The data also suggest that the population densities of *P. triticisoli* BJ-18 in the rhizosphere were much higher than in both the root and shoot.

Copy numbers of the 16S rRNA, *nifH* and ITS in the maize rhizosphere and root/shoot endosphere were determined by qPCR. As shown in Figure 2, copy numbers of the 16S rRNA gene per gram of rhizosphere soil, root and shoot were 10^9^, 10^5^ and 10^4^, respectively (Figure 2A–C). Whereas copy numbers of *nifH* gene in the rhizosphere, root and shoot were 10^4^, 10^2^ and 10^1^, respectively (Figure 2D–F). Inoculation significantly increased the copy numbers of both 16S rRNA and *nifH* under both low and medium N levels, but this effect was weakened under a high N level. Our findings that the rhizosphere had the highest bacterial and diazotrophic copy numbers among the three habitats support the observation that the roots are effective habitat filters, restricting the movement of bacteria from the soil to the roots [56]. We found that the copy numbers of ITS (fungi) in the rhizosphere were in the range of 4.96 × 10^6^ to 6.56 × 10^6^, and inoculation decreased the copy numbers of ITS under a low N level (Figure 2G).

### 2.2. Microbial Diversity

Samples of the maize rhizosphere, root and shoot were collected. MiSeq amplicon sequencing of each of the 16S rRNA, *nifH* gene and ITS region from 18 rhizosphere samples was carried out in order to characterize the rhizosphere bacterial, diazotrophic and fungal communities, respectively. Similarly, the 16S rRNA and *nifH* genes from 18 root and 18 shoot samples were MiSeq amplicon sequenced, respectively. The Good’s coverage (Appendix A) values were in the range of 0.974–0.978 (the rhizosphere bacteria), 0.998–0.999 (the root bacteria), 0.997–1.000 (the shoot bacteria), 0.994–0.996 (the rhizosphere diazotrophs), 0.986–0.993 (the root diazotrophs), 0.986–0.993 (the shoot diazotrophs) and 0.998–0.999 (the rhizosphere fungi), respectively. These data indicate that the current numbers of sequences were enough to reflect maize microbiome diversity. The rarefaction curves (Appendix A) also suggested that the microbial profiles were sufficient to represent the microbial communities.

Microbial community α-diversity, as evaluated by the Chao-1 and Shannon indices, is shown in Figure 3. The richness (Chao-1 indices) and diversity (Shannon indices) of bacteria, diazotrophs and fungi were calculated based on the rarefied sequences. Inoculation with *P. triticisoli* BJ-18 significantly reduced the bacterial Chao-1 richness and Shannon diversity in the rhizosphere under low N, medium N and high N levels (*t*-test, *p* < 0.05, hereafter), except Chao-1 under high N (Figure 3A). There were no significant effects on Chao-1 richness and Shannon diversity of the root endophytic bacteria between the inoculation and non-inoculation treatments under three N levels (Figure 3B). Inoculation significantly reduced the Shannon diversity of the shoot endophytic bacterial community (Figure 3C). Similarly, inoculation reduced the Chao-1 richness and Shannon diversity of diazotrophic communities in the rhizosphere, root and shoot, and significant levels were observed in low N (Figure 3D–F). There were no significant differences between the inoculated and uninoculated treatments in the fungal community of the rhizosphere (Figure 3G).

Microbial community β-diversity, as evaluated by principal coordinate analysis (PCoA), is shown in Figure 4. The PCoA and ANOSIM analyses clearly show that inoculation with *P. triticisoli* BJ-18 led to variation in the structure of the bacterial communities in the rhizosphere, root and shoot under both low N and medium N levels (Figure 4A–C). Similar results were obtained in the diazotrophic and fungal communities (Figure 4D–F).

### 2.3. Bacterial Communities in the Rhizosphere and Root/Shoot Endosphere

We found that there are obvious differences in the composition and diversity of the bacterial communities across the rhizosphere and root/shoot endosphere. No matter whether maize was inoculated with the *P. triticisoli* BJ-18 cells or not, the abundance and diversity of the bacterial phyla in the rhizosphere were much higher than in maize root/shoot endospheres (Figure 5A–C). The results are consistent with the observation that microbial population decreases from the rhizosphere to the endosphere [10]. In the non-inoculated treatments, Proteobacteria, Actinobacteria, Acidobacteria, Chloroflexi, Bacteroidetes, Firmicutes, Gemmatimonadetes, Planctomycetes, Nitrospirae and Cyanobacteria, in rank order, were the abundant phyla in the rhizosphere (Figure 5A), whereas Proteobacteria was mostly dominated in both root and shoot endospheres with the minor phyla Bacteroidetes, Actinobacteria and Firmicutes (Figure 5B,C). We found that inoculation with the *P. triticisoli* BJ-18 cells altered the structures of the bacterial microbiomes across all three compartments under both low N and medium N fertilizer conditions. Notably, inoculation enriched the Firmicutes, whose members include the *Bacillus* and *Paenibacillus* genera across all three compartments, especially in the shoot endosphere.

*Pseudomonas* (Proteobacteria phylum) was significantly dominated across three compartments. Notably, *Pseudarthrobacter* (Actinobacteria phylum) was the most abundant genus in the rhizosphere (Figure 5D), whereas *Pantoea* (Proteobacteria phylum) was mostly dominated in both root and shoot endospheres (Figure 5E,F). The data suggested that each habitat of the rhizosphere, root and shoot had its bacterial community, supporting that different plant organs were associated with different microbial communities [23]. Inoculation significantly increased the relative abundances of *Pseudomonas* (low N and medium N), *Salinimicrobium* (low N and medium N), Bacillus (low N, medium N, and high N), Lysobacter (low N and high N) and Devosia (low N) in the rhizosphere. Whereas Pseudomonas (low N) within root endosphere and Bacillus (low N and medium N) within shoot were mostly enriched, especially under low N level. It is well known that many members of *Pseudomonas* and *Bacillus* are PGPR used as biofertilizer or biocontrol agents.

### 2.4. Diazotrophic Communities in the Rhizosphere and Root/Shoot Endosphere

We analyzed the diazotrophic communities in the rhizosphere and root/shoot endosphere and found that there were obvious variations in the composition and diversity of the diazotrophic taxa across these compartments; also, inoculation significantly altered the abundance and diversity of diazotrophs across these compartments. The data were consistent with the results obtained in the bacterial communities among these compartments. The rhizosphere was significantly dominated by the phylum Proteobacteria, which was followed by Cyanobacteria, Verrucomicrobia and Firmicutes, whereas both root and shoot endospheres were mostly dominated by Proteobacteria (Figure 6A–C). Inoculation obviously increased the relative abundances of Proteobacteria (low N) in the rhizosphere, while inoculation significantly enriched Firmicutes within both the root (low N and medium N) and shoot (low N, medium N, and high N).

The dominant diazotrophs at the genus level in the rhizosphere mainly included *Klebsiella*, *Bradyrhizobium*, *Azotobacter*, *Azohydromonas*, *Skermanella*, *Trichormus* and *Azoarcus*, followed by *Paenibacillus*, *Leptothrix* and *Nostoc*, in rank order (Figure 6D). Whereas, both the root and shoot endospheres were dominated by both genera *Klebsiella* and *Paenibacillus* (~90% in relative abundance), followed by *Azotobacter*, *Skermanella*, *Azospirillum*, *Azoarus*, *Azohydromonas*, *Decloromonas*, *Bradyrhizobium* and *Methylobacter* (Figure 6E,F). Klebsiella, a member of Proteobacteria phylum, was the top genus dominated across all three compartments. It is well known that many members of *Klebsiella*, *Azotobacter*, *Azospirillum*, *Azoarus* and *Bradyrhizobium* are N_2_-fixing bacteria. Notably, inoculation significantly enriched *Klebsiella* (low N and medium N) in the rhizosphere. Whereas the relative abundance of *Paenibacillus* was significantly increased by inoculation within the root (low N and medium N) and shoot (low N, medium N, and high N). Other diazotrophs, such as *Azotobacter*, were also significantly enriched across all the compartments. The influences of inoculation on the composition of the diazotrophic communities were gradually weakened as the N level increased generally, consistent with the findings that the colonization rate of P. triticisoli BJ-18 was controlled by soil N status.

### 2.5. Fungal Community in the Maize Rhizosphere

In the rhizosphere fungal community, the phyla *Ascomycota*, *Zygomycota* and *Basidiomycota*, in rank order, were abundant (Figure 7A). At the genus level, the ten most abundant fungal genera were *Fusarium*, *Mortierlla*, *Gibberella*, *Talaromyces*, *Stachybotrys*, *Trichoderma*, *Peziza*, *Penicillium*, *Psilocybe* and *Humicola*, in rank order (Figure 7B). Inoculation significantly decreased the relative abundances of plant pathogen *Fusarium* (low N: *p* < 0.05) and *Gibberella* (low N: *p* < 0.01) under low N level.

### 2.6. Microbial Interaction Networks

The above results provide important insight into how a single taxon of a microbial community responds to *P. triticisoli* BJ-18 inoculation. Therefore, a multitude of direct and indirect interactions that occur in the bacterial, diazotrophic and fungal communities of the rhizosphere, root and shoot were further investigated (Figure 8, Figure 9 and Figure 10). In the rhizosphere, the bacterial, diazotrophic and fungal interactions changed little from low N to high N, as indicated by the connections between the nodes and links (Figure 8A,B, Figure 9A,B and Figure 10A,B). However, obvious changing patterns were observed in the root and shoot (Figure 8C–F, Figure 9C–F and Figure 10C–F). The bacterial interactions were stronger in the rhizosphere than those in the root and shoot (Figure 8), while the opposite results were observed in the diazotrophic communities (Figure 9). In the rhizosphere and endosphere, the potential interactions of the bacteria and diazotrophs in the *P. triticisoli* BJ-18-inoculated treatments were weaker than those in the uninoculated treatments, particularly in the low N level (Figure 8 and Figure 9). However, *P. triticisoli* BJ-18 inoculation promoted fungal interactions in the high N level (Figure 10). *P. triticisoli* BJ-18 inoculation reduced the rhizosphere bacterial and diazotrophic positive edge number ratios, while it increased the endosphere bacterial and diazotrophic and rhizosphere fungal positive edge number ratios.

### 2.7. Relationships between Microbial Community and Environmental Variables in the Rhizosphere and Root/Shoot Endosphere of Maize

The effects of inoculation with *P. triticisoli* on the soil properties and maize biomass and nutrition were investigated for the rhizosphere soil and maize seedling samples collected on Day 35 after planting. Nitrogenase activity of the rhizosphere was measured by using the acetylene reduction method, and the N_2_ fixation rate was performed by using the ^15^N_2_ incorporation assay. Compared to non-inoculation treatments, inoculation with *P. triticisoli* BJ-18 significantly increased the nitrogenase activities (Figure 11A) and N_2_ fixation rates (Figure 11B) under low N and medium N levels. Inoculation increased the soil pH, total N, organic matter and available P (Appendix A). Similarly, the N and P contents of maize seedlings were increased (Appendix A). After application of *P. triticisoli* BJ-18, the dry weights (biomass) of the root, shoot and total plant under all N levels were increased by 25.2%, 23.3% and 23.5%, respectively, under a low N level. Generally, the effects of inoculation on the soil and plant properties were most obvious under low N and more obvious under medium N than under high N.

To explore the relationships between microbial community structures and environmental factors (soil properties: pH, organic matter, total N, available P and plant properties: dry weight, N content and P content), a redundancy analysis (RDA) and Spearman’s correlation analysis were conducted. The RDA ordination showed that soil available P was the important factor in influencing the rhizosphere bacterial community, which was followed by soil total N (Figure 12A, Appendix A). Root dry weight was the key factor in influencing root bacterial community (Figure 12B, Appendix A). Whereas total N and total P were the major factors in influencing shoot bacterial community (Figure 12C, Appendix A). In the rhizosphere diazotrophic community, soil available P was the major influencing factor, followed by soil nitrogenase activity (Figure 12D, Appendix A). In the root endophytic diazotrophic community, root dry weight was the only significantly influencing factor (Figure 12E, Appendix A). In the shoot endophytic diazotrophic community, the influence factors were shoot total P, dry weight and total N (Figure 12F, Appendix A). In the rhizosphere fungal community, plant total dry weight was the major influencing factor, followed by soil total N and plant total P content (Figure 12G, Appendix A).

In the microbial communities of rhizosphere, the co-occurrence networks of environment–biology were further analyzed (Figure 13). Many bacterial genera in the rhizosphere, such as *Pseudomonas*, *Lysobacter* and *Bacillus*, were significantly and positively correlated with all or part of the soil properties (pH, available P, organic matter and total N) and plant properties (total dry weight and total P) (Figure 13A). In the fungal community, the abundances of *Gibberella* and *Fusarium* were significantly negatively correlated with soil nitrogenase activity, plant total N and plant total P (Figure 13B). In the diazotrophic community, the abundances of *Paenibacillus*, *Klebsiella*, and *Azotobacter* were significantly positively correlated with soil total N, soil available P and plant total dry weight (Figure 13C).

## 3. Discussion

In this study, the diazotrophic *P. triticisoli* BJ-18 was inoculated into maize grown in soil containing three different levels of N fertilizer. Then we examined the survival of the inoculant *P. triticisoli* BJ-18 and the effects of this inoculant on the composition and abundance of the bacterial and diazotrophic communities in all three compartments (rhizosphere, root and shoot), but fungi were only measured in the rhizosphere. All of the bacteria, diazotrophs and fungi in the rhizosphere were measured, since the rhizosphere is a large living site for microorganisms. The fungi were only measured, since the population of endophytic fungi in plant tissues is usually low. We found that the *P. triticisoli* BJ-18 cells survived and propagated in all habitats of the rhizosphere, root and shoot. Inoculation with *P. triticisoli* BJ-18 not only had influences on the soil and plant properties, but also significantly altered the structures of the bacterial, diazotrophic and fungal communities in all of these habitats. As far as we know, this is the first systematic study on the effects of the application of N_2_-fixing bacteria on the bacterial, diazotrophic and fungal communities in the rhizosphere and root/shoot endosphere of plants.

Firstly, we found that the *P. triticisoli* BJ-18 cells effectively colonized and propagated not only in the rhizosphere, but also in the maize root and shoot endospheres. The population densities of *P. triticisoli* BJ-18 across these compartments were significantly higher under low N than under high N, suggesting that the colonization of maize plants by this bacterium was controlled by the soil N status and consistent with the report that colonization of sugarcane by *Acetobacter diazotrophicus* was inhibited by high N fertilization [57]. Inoculation with *P. triticisoli* significantly increased the soil pH, total N, available P, organic matter, N_2_-fixation rate and nitrogenase activity. The N and P contents and biomass of the maize seedlings were also significantly enhanced by the inoculation. These good effects produced by inoculation of *P. triticisoli* BJ-18 are mainly due to the roles of *P. triticisoli* BJ-18 and the reshaped microbial community. *P. triticisoli* BJ-18 can provide N and Fe for plants by fixing nitrogen and by producing siderophores. We deduce that nitrogen fixation is more important in promoting plant growth than indolic acids, since *P. triticisoli* BJ-18 produces very low amounts of IAA [54]; also, we deduce that the altered microbial community play a very important role in improving soil and plant features. Especially, inoculation significantly decreased the relative abundances of the plant pathogen *Fusarium*. The antimicrobial compounds produced by *P. triticisoli* BJ-18 may be a contributor to the reduction in *Fusarium* density [54]. The enhanced populations of the diazotrophs and effective bacteria should be the contributors to improvement of the soil and plant properties. Our previous study showed that the ability of phosphate solubilization by *P. triticisoli* BJ-18 is very low [54]. The increased phosphorus in soil and plants may be due to the roles of phosphate solubilization by other effective microbes in the altered microbial community.

We comparatively analyzed the bacterial communities in the rhizosphere, root/shoot endosphere of maize seedlings grown in different N levels of fertilizer, with inoculation or without inoculation. No matter if the maize seedlings were inoculated or not, the abundance and diversity of the bacterial community in the rhizosphere were significantly higher than in root/shoot endospheres, supporting that the root provides an active and robust selection of bacteria for entering into plant tissues from the soil. Interestingly, the shoot endosphere was much higher than the root endosphere in diversity of the bacterial taxa, consistent with the reports that each plant tissue has its microbiome [10]. Notably, *Proteobacteria* was the dominant phylum across all the compartments, especially in root/shoot endospheres. In the non-inoculation treatments, the major bacterial phyla in the rhizosphere comprised Proteobacteria, Actinobacteria, Acidobacteria, Chloroflexi, Bacteroidetes, Firmicutes and Gemmatimonadetes. These major bacterial phyla observed in our study were also found in the bacterial community of the maize rhizosphere [38,58,59]. Our study revealed that the ten most dominant bacterial genera in the rhizosphere bacterial community were *Pseudarthrobacter*, *Pseudomonas*, *Nitrospira*, *Salinimicrobium*, *Bacillus*, *Lysobacter*, *Devosia*, *Gaiella*, *Nocardioides* and *Marmoricola*. Our results were a little different from the reports that *Chitinophaga*, *Flavisolibacter*, *Nitrospira*, *Pseudomonas* and *Streptomyces* were predominantly found in the maize rhizosphere and bulk soil [38]. We deduced that the difference in compositions of the bacterial communities may be due to soil type, which was one of the major factors in shaping the microbial community. Our findings that *Pseudarthrobacter*, a member of Actinobacteria phylum, was the dominant genus in the maize rhizosphere was also observed in beet soils [60]. Some members of the *Pseudarthrobacter* genus could degrade methane and benzoate [61]. Unlike in the rhizosphere bacterial community, *Pantoea* was the most abundant genus in both root and shoot bacterial communities. It was reported that *Pantoea* spp. as endophytes were carried by seeds of rice, wheat and crabgrass [62,63,64]. Some members of *Pantoea*, belonging to the PGPR, were used as an effective biocontrol agent, whereas *Pantoea stewartii* was a pathogen of maize [65,66,67]. Inoculation with *P. triticisoli* significantly enriched *Pseudomonas* in both the rhizosphere and root endosphere under all N levels, especially under low N and medium N levels. Whereas inoculation obviously increased the relative abundance of *Pseudomonas* in the shoot endosphere under high level and significantly enriched *Bacillus* in the shoot endosphere under low N and medium N levels. Many members of both the *Pseudomonas* and *Bacillus* genera as PGPR are widely used as biofertilizers or biocontrol agents [14,15]. The relative abundances of *Salinimicrobium*, *Lysobacter* and *Enterobacter* were also significantly increased by inoculation with *P. triticisoli*. Some members of *Lysobacter* were used as biocontrol agents against fungal diseases, since they could secrete antimicrobial compounds and extracellular enzymes against bacteria, fungi, oomycetes and nematodes [68,69]. We found that the abundances of *Pseudomonas* and *Bacillus* were significantly positively correlated with soil total N, soil available P, plant total P and plant biomass.

Although diazotrophs as PGPB are in common use as inoculants to improve crop yield and to reduce the consumption of chemical nitrogen fertilization, the studies on the diazotrophic community in the rhizosphere, especially in the root/shoot endosphere, are very few. Here, our study revealed that the abundance and diversity of the diazotrophic communities in the rhizosphere is much higher than in root/shoot endospheres, consistent with the observation in the bacterial community. However, the composition of the diazotrophic community structure across all the compartments was much simpler than that of the bacterial community structure, supporting that the diazotrophic community is only a small part of the bacterial community. Inoculation significantly enriched *Klebsiella* in the rhizosphere, some members (e.g., *Klebsiella oxytoca*) of which are well-known N_2_-fixing bacteria. Inoculation also significantly increased the relative abundance of the genus *Paenibacillus* across all the compartments, especially in the root/shoot endospheres, consistent with the observation that the inoculant *P. triticisoli* BJ-18 cells survived and propagated in all the habitats. In addition to *P. triticisoli* BJ-18, the diazotrophic *Paenibacillus* includes many members, such as *P. polymyxa*, *P. macerans*, *P. azotofixans*, *P. sabinae*, *P. zanthoxyli*, *P. forsythiae*, *P. sonchi*, *P. sophorae*, *P. jilunlii*, *P. taohuashanense* and *P. brasilensis* [53,70,71,72,73,74,75,76,77,78]. Certainly, the *P. triticisoli* BJ-18 cells were the main part of the *Paenibacillus* population in the inoculated rhizosphere and maize tissues. Other members, such as diazotrophic *P. polymyxa* and *P. brasilensismay*, also contribute to the enrichment of *Paenibacillus* across the compartments. Moreover, the other dominant genera, such as *Azospirillum*, *Azotobacter*, *Bradyrhizobium*, *Azoarcus*, *Azohydromonas* and *Skermanella*, were detected in the diazotrophic communities in the rhizosphere or in root/shoot endosphere. Our results are consistent with the reports that *Azohydromonas*, *Skermanella*, *Azotobacter* and *Bradyrhizobium* were the dominant genera in the diazotrophic community in a legume–oat intercropping soil [79]. Many members of these genera (e.g., *Azospirillum brasilense*, *Azotobacter chroococcum*) are used as biofertilizer to improve crop yield and to reduce the consumption of chemical nitrogen fertilization [80,81]. *Bradyrhizobium* is generally symbiotic with legumes, but some members of *Bradyrhizobium* are also free-living N_2_-fixers. *Bradyrhizobium* was also found in the rhizosphere of other non-legumes, such as maize, sugarcane and citrus [82,83,84]. We found that the diazotrophic communities across all the compartments included *Azoarcus*, which is also an important endophyte in rice [85,86]. Our data showed that the abundance of *Paenibacillus* and other diazotrophic genera, such as *Klebsiella*, *Azotobacter*, *Bradyrhizobium* and *Azoarcus*, were positively correlated with soil available P, nitrogenase activity, plant biomass and plant P and plant total N, suggesting that these diazotrophs are beneficial for plant growth. It is well known that some members of *Pseudomonas*, such as *P. stutzeri* A1501, are N_2_-fixng bacteria [41]. However, our study found that *Pseudomonas* was the main genus in the bacterial communities but was not dominant in the diazotrophic communities across all the compartments. The reason why *Pseudomonas* was not dominant in the diazotrophic communities may be due to the diazotrophic *Pseudomonas* species and strains forming only a very small part of the large *Pseudomonas* group.

Furthermore, the effects of inoculation with *P. triticisoli* BJ-18 on the rhizosphere fungal community were here investigated. Inoculation decreased the population density of fungi in the rhizosphere, consistent with the reports that application of *Bacillus amyloliquefaciens* NJN-6 decreased the fungal abundance in the rhizosphere soil of banana [87]. Especially, inoculation significantly decreased the relative abundances of *Fusarium* and *Gibberella* under a low N level, suggesting that *P. triticisoli* BJ-18 played an important role in inhibiting these plant pathogens. *Fusarium* and *Gibberella* are the anamorphic and teleomorphic state of the fungus, respectively [88]. *Fusarium graminearum* and *Fusarium moniliforme* caused *Gibberella* stalk rot, one of the most destructive soil-borne diseases of maize [89]. Many *Fusarium* species are pathogens of lots of plants, such as maize, wheat, cucumber and banana [88]. Some *Fusarium* species are not plant pathogens and even they are beneficial to plants. However, we could not distinguish which *Fusarium* species is pathogenic or beneficial in this study. The current study showed that the abundances of *Fusarium* and *Gibberella* were negatively correlated with soil available P, soil total N, plant total N, plant total P and plant biomass. Thus, we deduce that a reduction in the relative abundance of *Fusarium* and *Gibberella* in the maize rhizosphere by inoculation might be due to the antimicrobial substances produced by *P. triticisoli* BJ-18 [54]. In addition, other microbes, such as *Pseudomonas* and *Bacillus*, may also play roles in inhibiting plant pathogens, since these microorganisms were antagonistic microbes against *Fusarium* wilt [39,90].

## 4. Materials and Methods

### 4.1. Plant Plantation and Inoculation with P. triticisoli BJ-18 Cells

*P. triticisoli* BJ-18 cells were cultured in Luria Bertani broth medium [91] overnight at 30 °C, and then were harvested by centrifugation at 4000× *g* for 5 min, and adjusted to 10^8^ cells mL^−1^ with sterile normal saline solution. The concentration 10^8^ cell mL^−1^ was calculated by spectrophotometry and spread plate method.

The soil used in the pot experiments was low N-content sandy loam that was topsoil (0–20 cm depth) taken from the Shangzhuang Experimental Station of China Agricultural University, Beijing, China (40°08′12.15″ N, 116°10′44.83″ E, 50.21 m above sea level). The collected soil was air-dried at room temperature, and then was screened by a 10-mesh sieve to remove plant residues and reduce the heterogeneity of the soil. Before planting maize, P (50 mg Na_2_HPO_4_ per kg soil) and K (17 mg KCl per kg soil) were applied to the soil as base fertilizers. N fertilizers ((NH_4_)_2_SO_4_) were applied to the soil as base fertilizer at three N levels: high N (250 mg N kg^−1^ soil), medium N (166 mg N kg^−1^ soil) and low N (83 mg N kg^−1^ soil).

Maize seeds (*Zea mays* L., genotype hybrid Zhengdan 958 Henan Shangke Seed Co., Ltd., Shangqiu, China) were surface-sterilized with 10% sodium hypochlorite (NaClO) for 10 min, washed with sterile deionized water three times, and sprouted on sterile Petri dishes containing moist filter papers at room temperature (25 °C). After germination, uniform and vigorous seedlings were selected and divided into two groups at random. For the inoculated group (E+), the maize seedlings were soaked in the suspension (10^8^ cells mL^−1^) of *P. triticisoli* BJ-18 for 30 min to facilitate colonization. The non-inoculated group (E−) indicated that the maize seedlings were soaked in sterile saline solution. Then, the inoculated seedlings or non-inoculated seedlings were transplanted into square plastic pots (length of 55 cm; width of 30 cm; high of 17 cm) and 6 maize seedlings were planted in each per pot. On Day 7, 120 mL of the bacterial suspension (10^8^ cells mL^−1^) was applied to the pot containing the inoculated seedlings, and 120 mL of deionized water was applied to the pot containing the non-inoculated seedlings as the non-inoculated control. Each pot contained about 7.5 kg soil that was collected and supplemented with fertilizers of K, P and N as the base fertilizer, as described above. The seedlings were regularly watered (tap water) to 40% relative soil moisture by the weighing method [92] every 5 days. Three replications of each treatment were conducted in the greenhouse under optimum conditions (15 h light/9 h dark cycle and 25 °C–30 °C/15 °C–20 °C day/night temperature).

### 4.2. Sample Collection

For determination of survival and propagation of *P. triticisoli* BJ-18 in the maize rhizosphere and root/shoot endosphere, the maize seedlings were harvested from each treatment on Days 7, 12, 17, 27, 32 and 35 after transplanting, respectively. Whole maize seedlings were uprooted and separated into shoots and roots, both of which were then washed with deionized water to remove the adhering soil particles. In order to sterilize the outer surface of maize tissues, the roots and shoots were immersed in 70% ethanol for 30 s and subsequently in 2% NaClO solution added with Tween 80 (one droplet per 100 mL solution) for 10 min [93]. Subsequently, they were rinsed three times with sterile distilled water for 1 min. Finally, 0.1 mL of the final wash was spread on Luria Bertani agar plates to check whether the microorganisms were completely removed. These maize roots and shoots were immediately frozen in liquid N and then maintained at −80 °C for DNA extraction. At the same time, the tightly adhering soil on the maize roots was gathered as rhizosphere soil on Days 7, 12, 17, 27, 32 and 35 and then maintained at −80 °C.

The samples of the rhizosphere soil and maize root and shoot collected at Day 35 were used to extract DNA for analysis of the bacterial, diazotrophic and fungal communities. For assaying plant biomass (dry weight), the plant tissues were oven-dried at 105 °C for 30 min and then dried at 65 °C until constant weight.

### 4.3. Soil and Plant Physicochemical Property

Appropriate amount of soil and plant samples were digested by an H_2_SO_4_-H_2_O_2_ mixture at 370 °C, and then the N concentration was determined using the modified Kjeldahl method [94] and P concentration was determined using the standard method [95]. Available P was extracted with resin and measured according to the description [96]. The organic matter was measured according to the description [97]. Soil pH was measured using SevenMulti (Mettler-Toledo GmbH, Schwerzenbach, Switzerland) with soil:water of 1:1 [98].

### 4.4. Nitrogenase Activity Assay

Nitrogenase activity of the rhizosphere soil collected on Day 35 was measured by using the acetylene reduction method [99,100], with slight modification. Specifically, 2 g rhizosphere soil was added to a 26 mL glass jar fitted with septa for gas sampling and then sterile deionized water was added to 240% moisture content. After incubating the cultures for 24 h at 28 °C, with shaking at 180 rpm, the air in the glass jar was replaced with argon gas, and 10% (*v*/*v*) of the headspace argon gas in the tube was replaced with acetylene gas. After incubation for 2 h, 100 μL of gas from the tube was collected to measure the ethylene production using a GC-2010 Plus gas chromatograph (Shimadzu Corp., Kyoto, Japan).

### 4.5. ^15^N_2_ Incorporation Assay

A ^15^N_2_ incorporation assay was used to measure the incorporation of N_2_ gas into organic N by the N_2_-fixing microbes in the soil. Firstly, 2 g rhizosphere soil was added to a 26 mL glass jar fitted with septa for gas sampling and then sterile deionized water was added to 240% moisture content. After incubating the cultures for 24 h at 28 °C, with shaking at 180 rpm, the air in the glass jar was replaced with argon gas, and 10% (*v*/*v*) of the headspace argon gas in the tube was replaced with ^15^N_2_ (99%+, Shanghai Engineering Research Center for Stable Isotope, Shanghai, China). After 72 h of incubation at 28 °C, the soil was collected, oven-dried, ground, weighed and sealed into tin capsules. ^15^N enrichment (δ^15^N value) of soil was determined using a DELTA V Advantage isotope ratio mass spectrometer (Thermo Fisher Scientific, Inc., Waltham, MA, USA).

### 4.6. Extraction of DNA

Total DNA was extracted from the rhizosphere soil or plant tissue samples using the FastDNA^®^ SPIN Kit for soil (MP Biomedicals, Santa Ana, CA, USA), according to the manufacturer’s recommendation. DNA concentration and quality were measured using a spectrophotometer (Nanodrop 2000, Thermo Fisher Scientific, Waltham, MA, USA).

### 4.7. Quantitative-PCR (qPCR) Analysis

The qPCR method was used to quantify the abundance of total bacterial 16S rRNA [101], fungal ITS [101], total diazotrophic *nifH* [102] and *P. triticisoli* BJ-18 *nifH* gene. The primers used for the qPCR are listed in Appendix A. The fragments of 200 bp (bacteria), 300 bp (fungi), 450 bp (total diazotrophic bacteria) and 217 bp (*P. triticisoli* BJ-18) were amplified by conventional PCR, respectively. The products were ligated to PMD 19-T vector (TaKaRa, Otsu, Japan), and recombinant plasmids were obtained using a TIANprep Mini Plasmid Kit (Tiangen Biotech (Beijing) Co., Ltd., Beijing, China). Then the standard curve (R^2^ > 0.99) was generated with a dilution range of the recombinant plasmids from 1 × 10^1^ to 10 × 10^7^ copies. qPCR was performed using the 7500 Real-Time PCR detection system (Applied Biosystems, Foster City, CA, USA) and the program was as follows: 95 °C for 1 min, followed by 40 cycles of 94 °C for 15 s, 55 °C for 34 s and 72 °C for 15 s [103]. The 20 μL reaction mixture contained the SYBR^®^ Premix Ex Taq™ (Takara, Kyoto, Japan), primer pair, DNA and ddH_2_O. Each treatment had three biological replicates, with three technical replicates for each biological replicate. Gene copy numbers were calculated based on the standard curves. Each treatment had three biological replicates, with three technical replicates for each biological replicate.

### 4.8. Amplification and Sequencing of 16S rRNA, nifH Gene and ITS Region

The V4-V5 region of the bacterial 16S rRNA gene was amplified from the DNA isolated from the maize rhizosphere by using the primer pair 515F and 907R [104]. Primers 799F/1392R and 799F/1193R were used for analyzing the endophytic bacteria of the plant tissues [56,105,106]. The universal primers nifHF and nifHR for *nifH* genes were used to amplify the *nifH* gene from the rhizosphere DNA and plant genomic-DNA for analyzing the diazotrophic communities of the maize rhizosphere, root and shoot [107]. The primers ITS1F and ITS1R were used for amplification of the fungal ITS region from soil DNA [108]. PCR was carried out using ABI GeneAmp 9700 (Applied Biosystems, Foster City, CA, USA). Then amplicons were measured using QuantiFluor™-ST (Promega, Wisconsin, WI, USA), pooled in equimolar amounts and paired-end sequenced on an Illumina MiSeq platform (Illumina, San Diego, CA, USA) by Majorbio Co., Ltd. in Shanghai, China. Three independent biological replicates were provided for each treatment except for shoot endophytic bacteria (two independent biological replicates). These sequence data have been submitted to the GenBank databases under accession numbers SRP218893, SRP218883, SRP223202.

### 4.9. Bioinformatic Analysis

Raw reads of bacteria, diazotrophs and fungi were analyzed on the free online platform of Majorbio I-Sanger Cloud Platform (www.i-sanger.com). The sequences of the 16S rRNA gene were aligned against the Silva database, the sequences of the ITS against the Unite database and the sequences of the *nifH* gene against the functional gene database. Then they were analyzed at the phylum, class, order, family and genus levels by RDP Classifier [109]. The alpha (α) diversity indices of microbial communities were estimated by Mothur [110,111]. Shannon was used for estimating community diversity, Chao-1 for community richness and Coverage for sequencing depth. The differences between groups were performed in R with the ‘’stats” package. Principal coordinate analysis (PCoA) was calculated based on Bray–Curtis matrices and generated in R (RStudio, Inc., Boston, MA, USA) with the “ade4” package (v. 1.7.13) and ANOSIM analysis was performed in R with the “vegan” package (v. 2.5.4).

### 4.10. Statistical Analysis

The data of the soil physicochemical properties, plant biomass and nutrition, gene copies and α-diversity index were statistically analyzed for differences between groups by two-way analysis of variance in SPSS software version 20 (SPSS Inc., Chicago, IL, USA), and the results with *p* < 0.05 were considered to be statistically significant.

The interactions of “biology–biology” and “environment–biology” were visualized by a co-occurrence network [112]. In the network, a connected link denotes a significant (*p* < 0.05) and strong (0.6 < |r| <1) Spearman’s correlation between two variables. The size of each node is proportional to the number of connections (i.e., degree). The thickness of each link is proportional to the absolute value of Spearman’s correlation coefficient. The number of nodes and links, centralization of closeness/betweenness/degree and modularity were output with the igraph package in R [113]. The resulting networks were visualized using the interactive platform Gephi [114].

## 5. Conclusions

*P. triticisoli* BJ-18 inoculation altered the structures of the bacterial, diazotrophic and fungal communities in the maize rhizosphere, root and shoot. The bacterial and diazotrophic communities were specific in each compartment, and their abundance and diversity were significantly higher in the rhizosphere than in the root and shoot. Inoculation significantly enriched the Proteobacteria, Bacteroidetes and Firmicutes in the rhizosphere bacterial community, while inoculation obviously increased the relative abundance of the Firmicutes in both root and shoot bacterial communities. Compared to the non-inoculation treatments, the relative abundances of the genera *Paenibacillus*, *Pseudomonas* and *Bacillus* were significantly increased and the relative abundance of *Fusarium*/*Gibberella* was decreased by the inoculation. The abundance of *Paenibacillus* was significantly correlated with plant dry weight, nitrogenase, N_2_-fixing rate, P solubilization and other properties of the soil and plant. Our study provides insight into the plant growth-promotion mechanisms used by diazotrophs—not only through fixing N, but also by reshaping the microbiome structure.

## Figures and Tables

**Figure 1 ijms-22-01460-f001:**
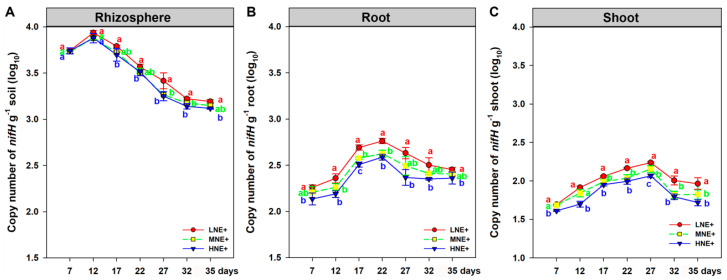
The population dynamics of *P. triticisoli* BJ-18 during the plant growth period from Days 7 to 35 in the maize rhizosphere (**A**), root (**B**) and shoot (**C**) during the plant growth period from Days 7 to 35. The population density is indicated by the copy number of the specific *nifH* gene determined by qPCR. The values are given as means of three independent biological replicates. Different letters (a, b and c) indicate significant differences between different N treatments at a certain day according to the LSD test (*p* < 0.05).

**Figure 2 ijms-22-01460-f002:**
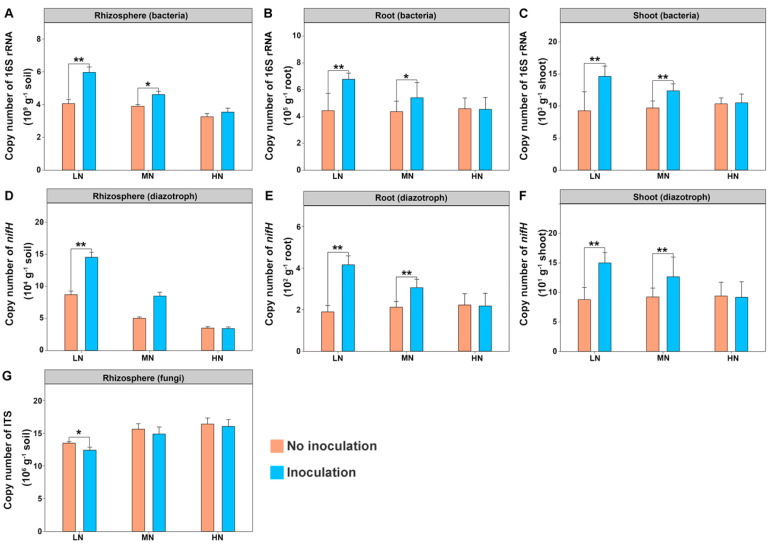
Copy numbers of the 16S rRNA gene (bacteria), *nifH* gene (diazotrophs) and ITS (fungi) in the maize rhizosphere, root and shoot: (**A**) the rhizosphere bacteria; (**B**) the root endophytic bacteria; (**C**) the shoot endophytic bacteria; (**D**) the rhizosphere diazotrophs; (**E**) the root endophytic diazotrophs; (**F**) the shoot endophytic diazotrophs; (**G**) the rhizosphere fungi. The values are given as the means of three independent biological replicates. The asterisk(s) (* or **) indicate significant differences between the inoculated and uninoculated groups determined by Student’s *t* at *p* < 0.05 or *p* < 0.01. LN: low nitrogen; MN: medium nitrogen; HN: high nitrogen.

**Figure 3 ijms-22-01460-f003:**
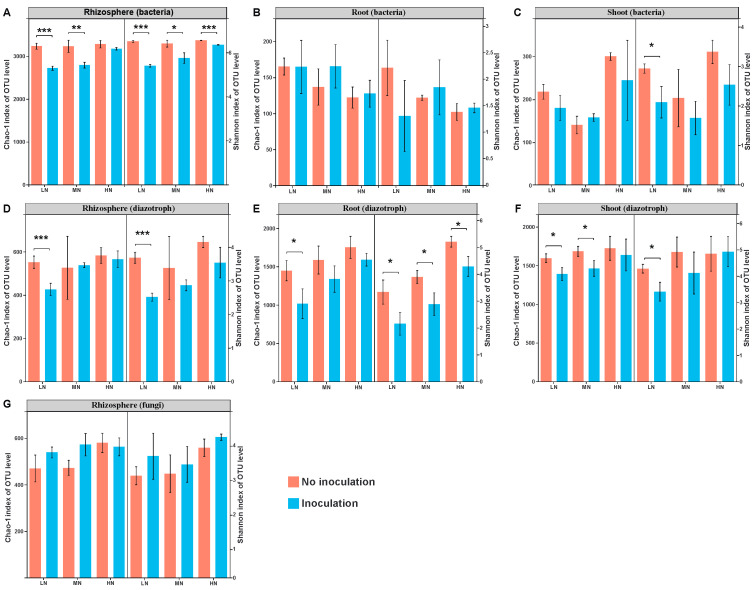
Alpha diversity of the bacterial, diazotrophic and fungal communities in the maize rhizosphere, root and shoot: (**A**) the rhizosphere bacteria; (**B**) the root endophytic bacteria; (**C**) the shoot endophytic bacteria; (**D**) the rhizosphere diazotrophs; (**E**) the root endophytic diazotrophs; (**F**) the shoot endophytic diazotrophs; (**G**) the rhizosphere fungi. The values are given as the means of three independent biological replicates except for shoot endophytic bacteria (two independent biological replicates). The asterisk(s) (* or ** or ***) indicate significant differences between the inoculated and uninoculated groups determined by Student’s *t* at *p* < 0.05 or *p* < 0.01 or *p* < 0.001. LN: low nitrogen; MN: medium nitrogen; HN: high nitrogen.

**Figure 4 ijms-22-01460-f004:**
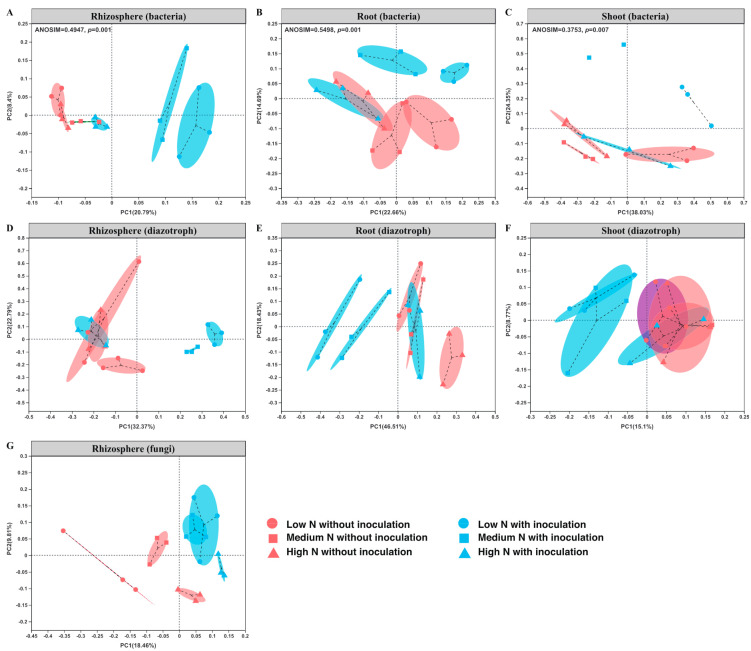
Principal coordinate analysis (PCoA) of the bacterial, diazotrophic and fungal communities in the maize rhizosphere, root and shoot: (**A**) the rhizosphere bacterial community; (**B**) the root endophytic bacterial community; (**C**) the shoot endophytic bacterial community; (**D**) the rhizosphere diazotrophic community; (**E**) the root endophytic diazotrophic community; (**F**) the shoot endophytic diazotrophic community; (**G**) the rhizosphere fungal community. Three independent biological replicates were provided for each treatment except for shoot endophytic bacteria (two independent biological replicates).

**Figure 5 ijms-22-01460-f005:**
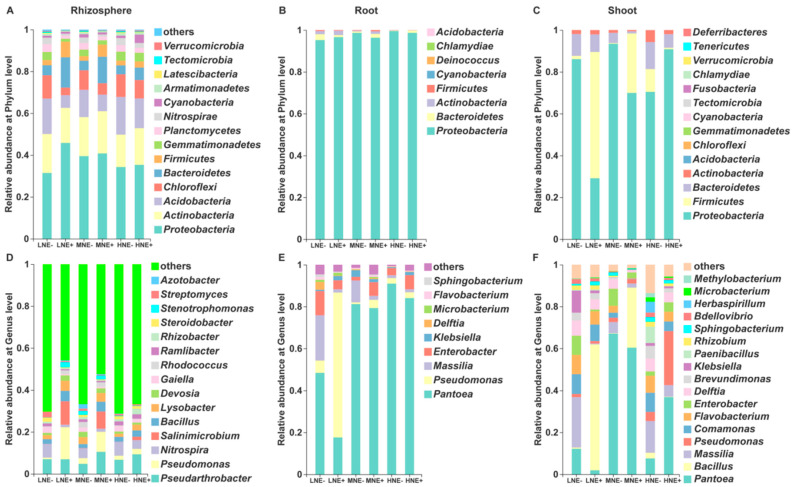
The composition of the bacterial communities in the rhizosphere and root/shoot endosphere at the phylum level (**A**–**C**) and genus level (**D**–**F**). LNE−: low nitrogen without inoculation; LNE+: low nitrogen with inoculation; MNE−: moderate nitrogen without inoculation; MNE+: moderate nitrogen with inoculation; HNE−: high nitrogen without inoculation; HNE+: high nitrogen with inoculation. The results were obtained based on 16s rRNA sequencing. The statistical comparison of the relative abundance of bacteria in the rhizosphere, roots and shoots at the genus level is shown in Appendix A.

**Figure 6 ijms-22-01460-f006:**
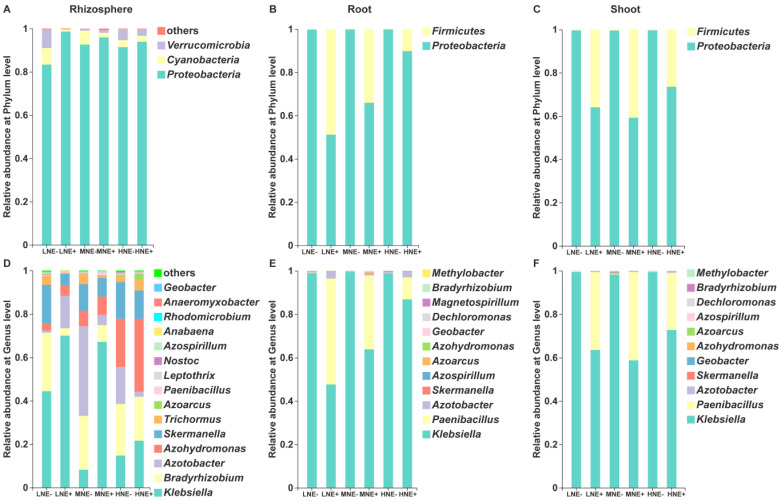
The composition of the diazotrophic communities in the rhizosphere and root/shoot endosphere at the phylum level (**A**–**C**) and genus level (**D–F**). LNE−: low nitrogen without inoculation; LNE+: low nitrogen with inoculation; MNE−: moderate nitrogen without inoculation; MNE+: moderate nitrogen with inoculation; HNE−: high nitrogen without inoculation; HNE+: high nitrogen with inoculation. The results were obtained based on the *nifH* sequencing. The statistical comparison of the relative abundance of diazotrophs in the rhizosphere, roots and shoots at the genus level is shown in Appendix A. The composition of the diazotrophic communities filtered out from all of the *Paenibacillus* reads are shown in Appendix A.

**Figure 7 ijms-22-01460-f007:**
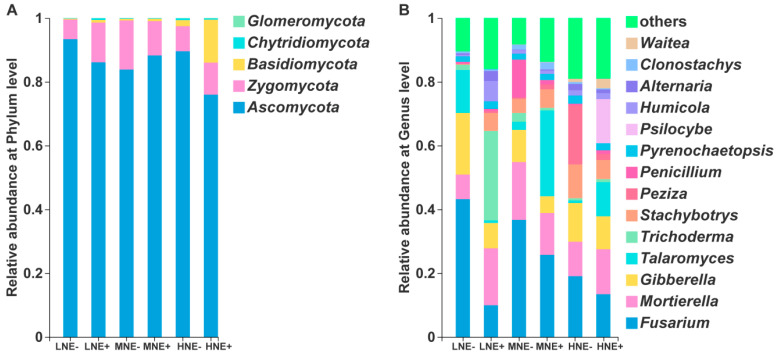
The composition of the fungal microbiome in the rhizosphere at the phylum level (**A**) and genus level (**B**). LNE−: low nitrogen without inoculation; LNE+: low nitrogen with inoculation; MNE−: moderate nitrogen without inoculation; MNE+: moderate nitrogen with inoculation; HNE−: high nitrogen without inoculation; HNE+: high nitrogen with inoculation. The results were obtained based on ITS sequencing. The statistical comparison of the relative abundance of fungi in the rhizosphere at the genus level is shown in Appendix A.

**Figure 8 ijms-22-01460-f008:**
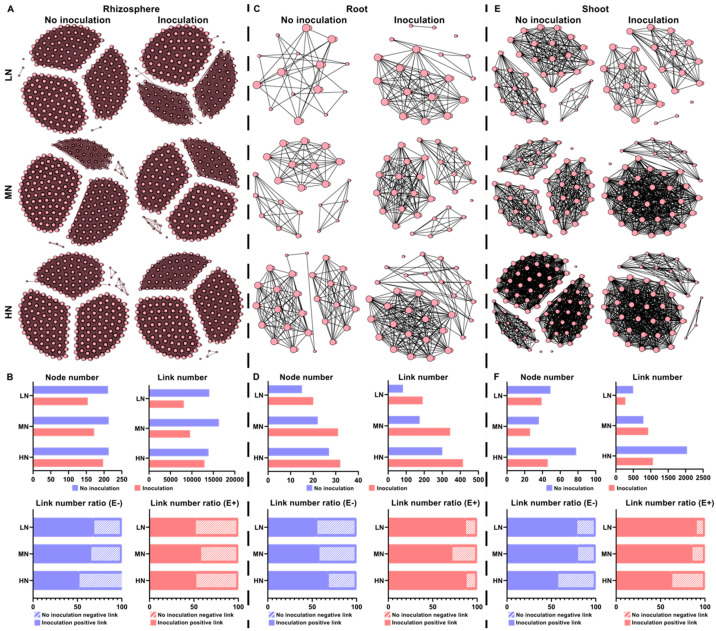
Co-occurrence network of biology–biology, representing the bacterial interactions in the rhizosphere (**A**), root (**C**) and shoot (**E**) at different nitrogen levels. The topological characterizations include the node number, link number and link number ratio (positive link versus negative link; (**B**,**D**,**F**)). The connections are Spearman correlations with significance (*p* < 0.05) and high significance (0.6 < |r| < 1).

**Figure 9 ijms-22-01460-f009:**
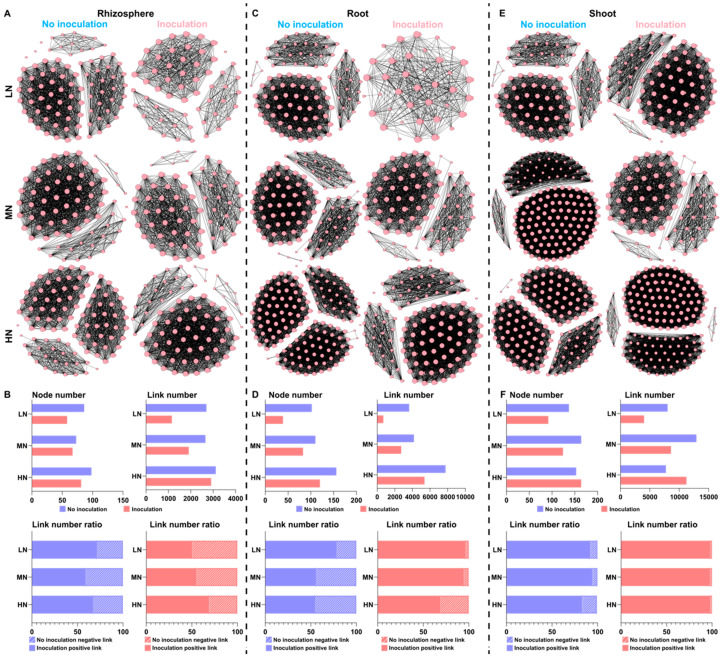
Co-occurrence network of biology–biology, representing the diazotrophic interactions in the rhizosphere (**A**), root (**C**) and shoot (**E**) at different nitrogen levels. The topological characterizations include the node number, link number and link number ratio (positive link versus negative link; (**B**,**D**,**F**)). The connections are Spearman correlations with significance (*p* < 0.05) and high significance (0.6 < |r| < 1).

**Figure 10 ijms-22-01460-f010:**
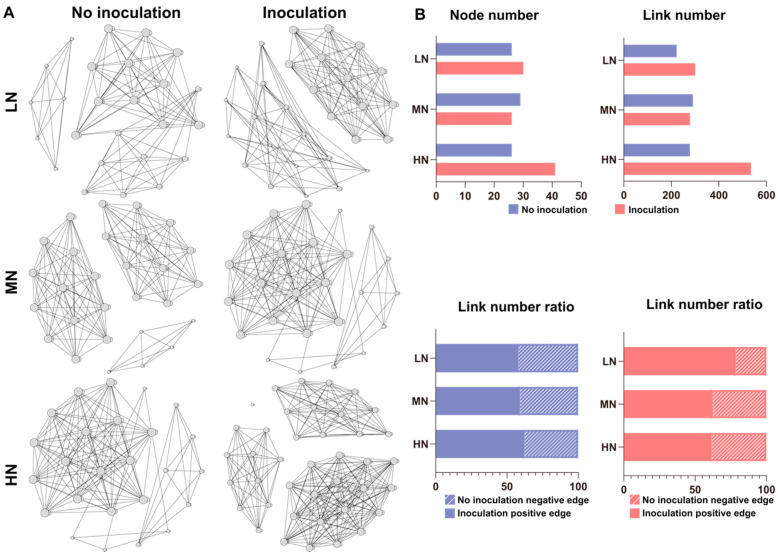
Co-occurrence network of biology–biology, representing the fungal interactions in the rhizosphere at different nitrogen levels (**A**). The topological characterizations include the node number, link number and link number ratio (positive link versus negative link) (**B**). The connections are Spearman correlations with significance (*p* < 0.05) and high significance (0.6 < |r| < 1).

**Figure 11 ijms-22-01460-f011:**
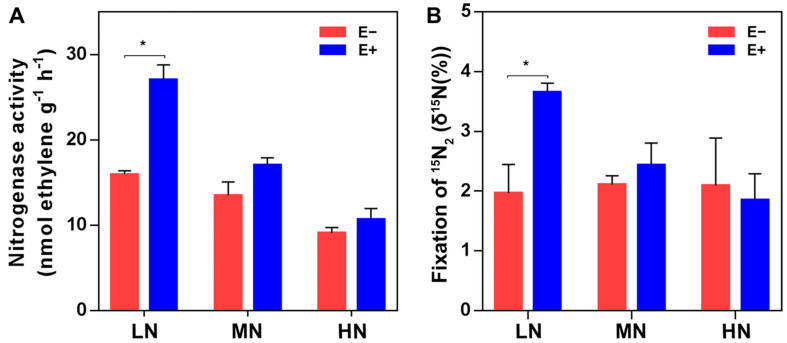
The nitrogen-fixing ability of maize rhizosphere soil. Nitrogenase activity was determined with an acetylene reduction assay (**A**) and ^15^N_2_ incorporation assay (**B**). The values are given as the means of three independent biological replicates. The asterisk (*) indicate significant differences between the inoculated (E+) and non-inoculated (E−) groups, determined by Student’s *t* at *p* < 0.05.

**Figure 12 ijms-22-01460-f012:**
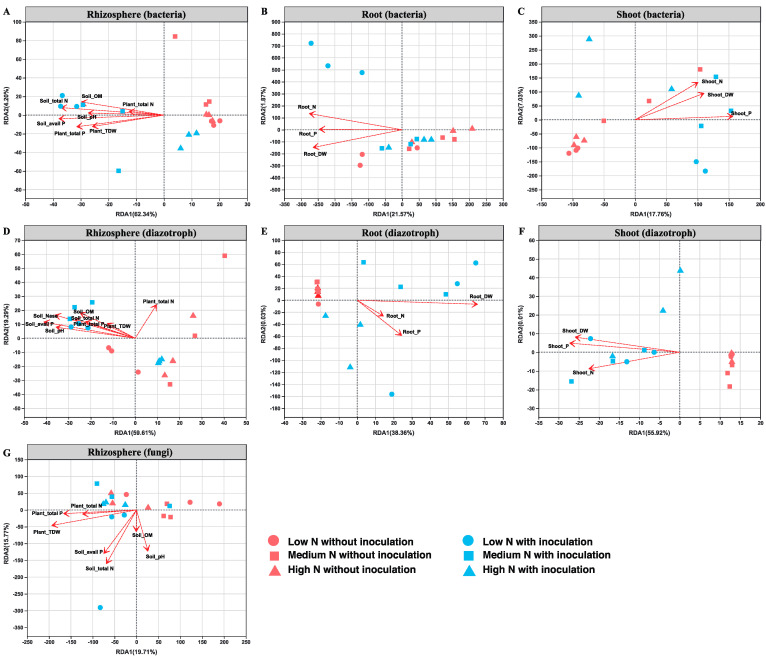
Redundancy analysis (RDA) of the bacterial, diazotrophic and fungal communities in the maize rhizosphere and root/shoot endosphere: (**A**) the rhizosphere bacteria; (**B**) the root endophytic bacteria; (**C**) the shoot endophytic bacteria; (**D**) the rhizosphere diazotrophs; (**E**) the root endophytic diazotrophs; (**F**) the shoot endophytic diazotrophs; (**G**) the rhizosphere fungi. Soil_total N: soil total nitrogen; Soil_avail P: soil available phosphorous; Soil_OM: soil organic matter; Soil_Nase: soil nitrogenase activity; Plant_TDW: plant total dry weight; Plant_total N: plant total nitrogen content; Plant_total P: plant total phosphorous content; Root_DW: root dry weigh; Root_N: root nitrogen content; Root_P: root phosphorous content; Shoot_DW: shoot dry weight; Shoot _N: shoot nitrogen content; Shoot _P: shoot phosphorous content.

**Figure 13 ijms-22-01460-f013:**
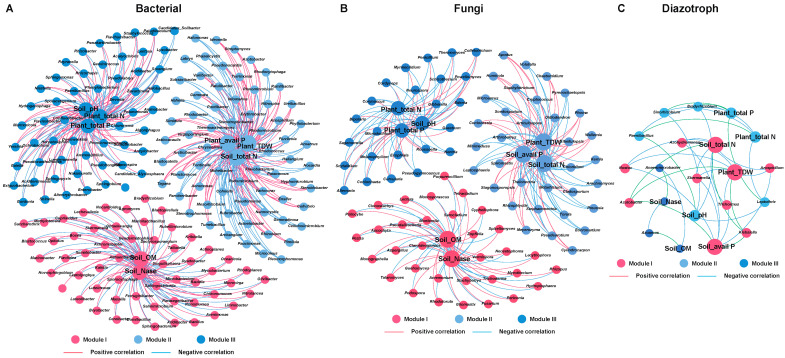
Co-occurrence network of environment–biology, representing the correlations between the environment factors and the relative abundances of the bacteria (**A**), fungi (**B**) and diazotrophs (**C**). The connections are Spearman correlations with significance (*p* < 0.05) and high significance (0.6 < |r| < 1, red line: positive correlation, blue line: negative correlation). The network is color coded by module, which means that the nodes clustered in the same module share the same color.

## Data Availability

The DNA reads have been deposited at NCBI under the SRA accession no. SRP218893, SRP218883, SRP223202.

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
