# Peer review of "Diazotroph *Paenibacillus triticisoli* BJ-18 Drives the Variation in Bacterial, Diazotrophic and Fungal Communities in the Rhizosphere and Root/Shoot Endosphere of Maize"

_ijms, 2021, doi:10.3390/ijms22031460_

Round 1

Reviewer 1 Report

Comments or Suggestions for Authors

I found 2 major issues, 3 moderate issues, and several minor issues with your manuscript. The good news is that I do not see the need to redo any wet lab experiments. (I cannot speak for the editor or any other reviewers, though.) Some bioinformatic or statistical analyses may need to be rerun, though. Even so, I think that the requested changes can be made relatively quickly and the manuscript can be resubmitted soon.

Major Issues

Issue 1—Confusing Figures and Tables

Many of your figures and tables need to be adjusted so that they are easier to interpret and understand.

I think that Figure 1 (p. 4, lines 149–160) would be easier to interpret with a few adjustments. First, set the y-axis for all three panels to be the same (1.5 to 4.0) so that compartments can be more easily compared. Second, the key at the bottom is more eye-grabbing than the titles, so it's easy to think that A, B, and C are "low N, medium N, and high N" instead of "Rhizosphere, Root, and Shoot". To fix this make the key smaller and put a copy inside each panel (A, B, and C). Third, having red circles and green circles may be confusing to individuals who are color blind. Consider changing the green circles to yellow squares. Fourth, make the color of the error bars correspond to the color of the line they describe (e.g., red error bars on the red lines). Finally, consider using colors to indicate which letters correspond to which lines (e.g., blue a, b, and c for the blue line).

In Table 1 (p. 5, lines 181–187), it's not clear to me which numbers the * and ** are associated with. In my copy of the .pdf, at least, they are sitting in their own row. So do they refer to the numbers above them or the numbers below them? This needs to be fixed. And consider turning this into a single bar graph.

Figure 2 (p. 6, line 212) needs a key to explain what light blue, dark blue, light orange, and dark orange means. The figure caption should also explain what the colors mean and it should also explain that the left side of each plot is the Chao1 index and the right side of each plot is the Shannon index. And the caption should say how many biological replicates are represented. If it's only three biological replicates, then box plots are not appropriate. You should represent the three data points with individual dots.

Figure 3 (p. 7, lines 229–244): Even though it's obvious in the figure, the caption should say how many biological replicates are represented. Are there only two replicates for "Medium N with inoculation" in Figure 3C? If so, why?

Figure 4 (p. 8, lines 278–283), Figure 5 (p. 9, lines 310–315), and Figure 6 (p. 10, lines 325–329): These are stacked bar charts, not histograms. The captions should say how many biological replicates are represented. Also, consider adding error bars so that the reader can get an idea of how much variability there was between samples for each phylum or genus. If possible, be consistent in your use of colors. For example, in Figure 5, pale turquoise (#8dd3c7) represents Proteobacteria in A, B, and C. But light yellow (#ffffb3) represents Cyanobacteria in A and Firmicutes in B and C. It would be easier if light yellow represented Cyanobacteria in A and light blue (#80b0d3) represented Firmicutes in B and C. This would also allow you to use two color keys instead of six: one for Figures 5A–C and another for Figures 5D–F. You mention a lot of the taxa from these graphs in the results, but you do not report whether these are statistically significant. There's not a good way to report significance in the graphs, but there is no reason you can't report those numbers in the text. Also, I would be curious to see what Figures 5D–E look like if you filter out all of the Paenibacillus reads. Perhaps this could be included as another supplemental figure.

Figure 7 (p. 11), Figure 8 (p. 12), and Figure 9 (p. 13, lines 367–371): These are really complicated figures; both the text (p. 13, lines 330–345) and the captions do very little to help the reader know how to understand them. You need to elaborate on these figures substantially. What are the pink nodes? Pink edges? Blue nodes? Blue edges? Also, when networks are presented as "hairballs", they tend to be uninformative and brittle (see DOI 10.1093/bib/bbr069). Also, it looks like the rotations of your networks are arbitrary; this implies information where there is none. Consider using hive plots instead and explain them to your readers; otherwise you have lots of pretty but uninformative pictures. Also, use "No inoculation" instead of the non-word "Uninoculation". Are there any statistically significant differences in the bar plots in B, D, and F for each of these graphs?

Figure 11 (p. 15, lines 413–422): The key given in the figure caption does not correspond to what is shown in the graphs. For example, Figure 11A has a label that says "OM", but the label described in the caption is "Soil_OM". The caption should say, "OM". Also, the discussion the text doesn't seem to match the figure, either. For example, on p. 14, lines 401–402, the text says, "Root dry weight was the key factor in influencing root bacterial community (Figure 11B, Supplementary Table S6)." But Figure 11B shows A, B, and N; I'm not sure what any of those are, but clearly none of them are RDW (root dry weight). Similar problems exist for the rest of the paragraph (p. 14, lines 396–410).

Figure 12 (p. 16, lines 433–438): The figure shows three different "models" for each figure but the caption says the figures are organized into three different "modules". Which is it? And how were they chosen? Also, it's not clear to me why the three models/modules are completely separated in A and B, but they are all mixed together in C. Furthermore, am I really to understand that there is no significant correlation between Soil_total N (Model/Module II) and Plant_total N (Model/Module III)? The text says, "In the diazotrophic community, the abundances of Paenibacillus, Klebsiella, and Azotobacter were significantly positively correlated with soil total N, soil available P, and plant total dry weight (Figure 12C)." (p. 15, lines 429–431) I'd like to be able to verify that, but the figure is so busy and the labels are so tiny, that I can't make out anything useful at all.

Issue 2—Mixed-up or Missing References

Some of your in-line citations are incorrect. Here are two examples (p. 3, lines 93–101):

  • "The nif gene transcription of polymyxa WLY78 was strongly regulated by NH4+ and O2 [20]." But reference 20 (Bai et al. Functional overlap of the Arabidopsis leaf and root microbiota. Nature 2015, 528, 364–370) says nothing about P. polymyxa.
  • "This bacterium exhibited the highest nitrogenase activity in the absence of NH4+ and no activity in the presence of more than 5 mM NH4+ under anaerobic conditions. Recent studies have revealed that GlnR (a central regulator of nitrogen metabolism) activates nif gene transcription under nitrogen limitation, whereas GlnR, together with glutamine synthetase (GS) encoded by glnA within glnRA operon, represses nif expression under excess nitrogen [21]." But reference 21 (Bulgari et al. Endophytic bacterial community of grapevine leaves influenced by sampling date and phytoplasma infection process. BMC Microbiology 2014, 14, 198) says nothing about glutamine synthetase.

I didn't check them all, so you will need to double-check all of your references.

Moderate Issues

Choice of Compartments

You measured bacteria and diazotrophic bacteria in all three compartments (rhizosphere, root, and shoot), but fungi were only measured in the rhizosphere. You never explain why. If you never measured ITS copies for the root and shoot compartments, you need to explain why you made that choice. If you did measure ITS copies for the root and shoot compartments, but had to exclude that data from the analysis, you need to explain why.

Where is Paenibacillus?

Figure 4 (p. 8, lines 278–283) shows phylum and genus abundance of bacteria for the different treatments. Remarkably, Paenibacillus only appears in Figure 4F and is only easily visible for one treatment (HNE−). Paenibacillus is there, just in very low abundance as evidenced by Figure 5 (p. 9, lines 310–315). Is it reasonable that a low-abundance member of the community has such large effects on the community structure as a whole? I think you need to confront this question in your discussion.

Additional Discussion Items

I'd like to see the following addressed in your discussion:

In the introduction you mention that P. triticisoli promotes plant growth through three mechanisms: iron-scavening siderophores, phytohormone production, and nitrogen fixation (p. 3, lines 120–126). You should revisit this in your discussion. How much of your data can be explained by nitrogen fixation and how much might be due to siderophores and/or phytohormones?

Also, in the introduction you mention the enormous role that plant root exudates play in shaping the plant rhizosphere microbiome (p. 2, lines 62–89). How much of your data can be explained directly by inoculation with P. triticisoli and how much of it is explained indirectly because the inoculated plants produced more root exudates?

You also find that phosphorus is a major driver of bacterial diversity in and around maize (p. 14, lines 396–410). You don't have any evidence of phosphate solubilization by P. triticisoli and the soil in the greenhouse pots should have started with roughly equivalent levels of available phosphorus. So why is phosphorus playing such a huge role?

Minor Issues

Overall, the English is generally very good. However, there are still a few grammatical mistakes found throughout the manuscript.

  1. 2, lines 43–45: You say, "Many rhizobacteria are Plant-Growth Promoting Rhizobacteria (PGPR) that improve plant growth by direct mechanisms (e.g. N fixation, phosphate solubilization, sequestering iron) and indirect mechanisms (e.g. indole-3-acetic acid, cytokinins, gibberellins)[6,7]." However, that's not what your sources say. Glick [your reference 6] says, "PGPB may promote plant growth directly usually by either facilitating resource acquisition or modulating plant hormone levels, or indirectly by decreasing the inhibitory effects of various pathogenic agents on plant growth and development, that is, by acting as biocontrol bacteria." Etesamia & Maheshwarib [your reference 7] do not define direct and indirect mechanisms of plant growth promotion (though they do mention them). Therefore, you should bring your statement into conformity with Glick's schema.
  2. 2, lines 57–59: "The number of bacterial cells in bulk soil is about 106-109 bacterial cells/g soil and in rhizosphere is about 106-109 bacterial cells/g soil), while the number of bacterial cells within root endosphere is about 104-108 per gram of root tissues." When using scientific notation, powers should be superscripted, like this: " The number of bacterial cells in bulk soil is about 106–109 bacterial cells/g soil and in rhizosphere is about 106–109 bacterial cells/g soil), while the number of bacterial cells within root endosphere is about 104–108 per gram of root tissues." Other cases where you need to use a superscript: p. 4, lines 144–147, 168–170, 174–176; p. 18, lines 552, 566; p. 19, line 570; and p. 20, line 633. It's possible I missed some, so check everything over.
  3. 2, lines 68: "Rhizobia" doesn't need to be capitalized here.
  4. 3, line 103: Since you're referring to the bacterial lifestyle, not the genus, you should use "rhizobia", not "Rhizobia" here.
  5. 5, line 182 and many other places (including supplemental files): Instead of using a hyphen (-) for LNE-and MNE- and HNE-, you should consider using a minus symbol (−): LNE−, MNE−, and HNE−. It looks much better.
  6. 5, lines 200–201: "As Microbial community α-diversity, as evaluated by Chao-1 and Shannon indices, was shown in Figure 2." should be "Microbial community α-diversity, as evaluated by Chao1 and Shannon indices, is shown in Figure 2."
  7. 13, line 376: Use "planting" instead of "plantation".
  8. 19, line 574: Was the tap water sterilized before using it for watering?
  9. 19, line 595: I've never done these experiments myself, so I could be wrong, but it looks like "370°C" might be a typo and it should really say, "37 °C".

In the references, several scientific names need to be capitalized, e.g., Paenibacillus polymyxa (p. 22, line 739), Azoarcus (p. 24, line 800), etc. I did not look for all of them; you will have to check for them.

Supplemental Table S1: It think "nifH specific for P. beijingensis BJ-18" should be "nifH specific for P. triticisoli BJ-18".

Supplemental Table S3: Make the letters superscript so that they're easier to assess. E.g., "6.83 ± 0.12bc" should be "6.83 ± 0.12bc". Also, you should indicate whether the soil was tested before planting and confirmed to be statistically uniform across these measurements. Otherwise, you must admit the possibility that the heterogeneity that you measured was already present before you planted and inoculated. Finally, you tested for the presence of nitrogen; you did not measure nitrogenase activity. So the title for this table should just be "Soil physicochemical properties."

Supplemental Table S4: Make the letters superscript so that they're easier to assess. E.g., "23.7 ± 0.1c" should be "23.7 ± 0.1c".

Supplemental Table S5 (and elsewhere): Instead of using three hyphens (---) to indicate missing or omitted data, consider using an em dash (—). It looks much better.

Reviewer 2 Report

In the manuscript “ Diazotroph Paenibacillus triticisoli BJ-18 drives variation of the bacterial, diazotrophic and fungal communities in the rhizosphere and root/shoot endosphere of maize” the authors investigate the survival/propagation of P. triticisoli BJ-18 and its effect on the maize rhizosphere, root/shoot endosphere.  P. triticisoli BJ-18 not only had influences on soil and plant properties, but also reshaped the structures of the bacterial, diazotrophic and fungal communities.

In general, I am rather positive about this work. The authors made an interesting story to read. However, there are several issues that have to be attended. Finally, the manuscript will benefits of a proper English editing.

Here the minor and minor comments.

Abstract

 Please can you improve the sentences from line 29 to 32?

Introduction

At line 52 delete “et al.”.

At line 108 change "plant root and shoot" in "root and shoot tissues"

Please improve the sentences form line 122 to 126. It seems that the two sentences are referred to two different research works while actually you are citing the same reference (number 18).

Results

At line 140-141: delete “but it was not found in the” with “than” and delete “of P. tritisicoli

At line 163 delete “in P. tritisicoli BJ18” and insert “determined by pPCR”.

At line 165-266: “nitrogenase activity of P. sabinae T27 grown anaerobically in medium containing different concentration of NH4Cl”. Is the sentence correct? I didn’t find any mention in the Material and Methods.

In table 1 decrease the font size.

At line 189 delete both “In this study” and “Then, we carried out” changing the sentence in “Miseq Amplicon sequencing of 16rRNA, nifH and ITS genes from 18 rhizosphere samples was carried out in order to characterize the rhizosphere bacterial, diazotrophic and fungal communities, respectively

The sentences at line 251-253 and 256-257 are very similar to the sentences from line 285 to 288. Please could the authors modify and improve them also using different terms as obvious (check also at line 292,384,385) and across/among these compartments.

At line 279, 311 and 326 delete “The histogram of “ and insert at the end of figure 4, 5 and 6 which method/analysis did you use to obtain that result. Also in the figure 6 description insert that the results are referred to rhizosfere compartment.

I guess that the sentence from line 322 to 324 has to be moved in the discussion section.

At line 374 delete “In this study”. In general try to delete it in the results section because you use it a lot of time in the discussion.

Materials and Methods

At line 550: change “inoculation with the cells of P. triticisoli BJ-18” in “inoculation with P. triticisoli BJ-18 cells”

At line 551: change “Cells of P. triticisoli BJ-18” in “P. triticisoli BJ-18 cells” and check in all the manuscript. Insert a reference concerning the LB composition.

At line 552-553: Please describe how did you measure 108 cell ml-1 and insert the concentration of saline solution.

At line 554 please delete “Pot experiments were conducted in the greenhouse” because you mention the environmental conditions at the end of the paragraph.

At line 557-558: “Before planting maize, P (Na2HPO4) and K (KCl) were applied to soil as base fertilizers at amounts of 50 mg P per kg soil and 17 mg K per kg soil ” please change it or modify in “Before planting maize, P (50 mg Na2HPO4 per kg soil ) and K (17 mg KCl per kg soil) were applied to soil as base fertilizers

At line 558: Change “(NH4)2SO4)” in  “((NH4)2SO4)” 

At line 561 change Zea mays in italics form Zea mays

At line 562 please insert the chemical formula of sodium hypochlorite

At line 567 please insert “sterile saline solution”

At line 574 please insert the reference concerning the determination of relative soil moisture by weighing method.

At line 579-580,137,: Change “in the maize rhizosphere, root endosphere and shoot endosphere” in “in the maize rhizosphere, root/shoot endosphere”. Check in all the manuscript.

At line 592: Change “the maize seedlings roots and shoots” in “ the plant tissues

At line 637: Is it correct “techigh Nical”?

At line 658 there is a space “Mothur   [108,109]

Conclusions

At line 676:Change “Inoculation with P. triticisoli BJ-18“ in “P. triticisoli BJ-18 inoculation

From line 676 to 680 please insert the plant Zea mays and in general improve the structure of sentences there are repetitions.

Insert a short description for supplementary Figure 1.
